# Beneficial Effects of Adiponectin on Glucose and Lipid Metabolism and Atherosclerotic Progression: Mechanisms and Perspectives

**DOI:** 10.3390/ijms20051190

**Published:** 2019-03-08

**Authors:** Hidekatsu Yanai, Hiroshi Yoshida

**Affiliations:** 1Department of Internal Medicine, National Center for Global Health and Medicine Kohnodai Hospital, 1-7-1 Kohnodai, Chiba 272-8516, Japan; 2Department of Laboratory Medicine, The Jikei University Kashiwa Hospital, 163-1 Kashiwashita, Kashiwa, Chiba 277-8567, Japan

**Keywords:** adiponectin, atherosclerosis, cholesterol efflux, diabetes, inflammation

## Abstract

Circulating adiponectin concentrations are reduced in obese individuals, and this reduction has been proposed to have a crucial role in the pathogenesis of atherosclerosis and cardiovascular diseases associated with obesity and the metabolic syndrome. We focus on the effects of adiponectin on glucose and lipid metabolism and on the molecular anti-atherosclerotic properties of adiponectin and also discuss the factors that increase the circulating levels of adiponectin. Adiponectin reduces inflammatory cytokines and oxidative stress, which leads to an improvement of insulin resistance. Adiponectin-induced improvement of insulin resistance and adiponectin itself reduce hepatic glucose production and increase the utilization of glucose and fatty acids by skeletal muscles, lowering blood glucose levels. Adiponectin has also β cell protective effects and may prevent the development of diabetes. Adiponectin concentration has been found to be correlated with lipoprotein metabolism; especially, it is associated with the metabolism of high-density lipoprotein (HDL) and triglyceride (TG). Adiponectin appears to increase HDL and decrease TG. Adiponectin increases ATP-binding cassette transporter A1 and lipoprotein lipase (LPL) and decreases hepatic lipase, which may elevate HDL. Increased LPL mass/activity and very low density lipoprotein (VLDL) receptor and reduced apo-CIII may increase VLDL catabolism and result in the reduction of serum TG. Further, adiponectin has various molecular anti-atherosclerotic properties, such as reduction of scavenger receptors in macrophages and increase of cholesterol efflux. These findings suggest that high levels of circulating adiponectin can protect against atherosclerosis. Weight loss, exercise, nutritional factors, anti-diabetic drugs, lipid-lowering drugs, and anti-hypertensive drugs have been associated with an increase of serum adiponectin level.

## 1. Introduction

Previously, the adipose tissue was considered a generally passive repository for stored triglycerides (TG). With the discovery of adiponectin, it has become clear that the adipose tissue carries out a large number of intricate metabolic, paracrine, and endocrine functions. The adiponectin gene was found to be the most abundantly expressed gene in the adipose tissue. It encodes a 244-amino-acid protein with a predicted size of 30 kDa [1]. Adiponectin contains a putative N-terminal signal sequence and a collagen-like domain and structurally belongs to the complement 1q (C1q) family, being characterized by a carboxyl-terminal globular domain and an amino-terminal collagenous domain highly homologous to collagen X, VIII, and C1q. Adiponectin is exclusively expressed and secreted into the circulation by the adipose tissue and appears to act as a hormone which could reduce inflammatory responses in vitro [2,3]. 

Circulating adiponectin can exist as a trimer, hexamer, or higher-order multimer with 12–18 subunits [4,5]. Adiponectin receptors include two similar transmembrane receptors which are known as AdipoR1 and AdipoR2, and another type of receptor without a transmembrane domain which may act as a co-receptor for the high-molecular weight (HMW) form of adiponectin on endothelial and smooth muscle cells [6,7]. Recent data indicate that the HMW form has the predominant action in metabolic tissues [8]. Adiponectin accounts for about 0.01% of all plasma proteins (5–10 μg/mL), and its plasma concentration was reported to be higher in women than in men [9,10].

Circulating adiponectin concentrations are reduced in obese individuals [10], and this reduction was proposed to have a crucial role in the pathogenesis of atherosclerosis and cardiovascular diseases associated with obesity and the metabolic syndrome [11,12]. Furthermore, this idea is supported by reports that adiponectin has effects considered to be protective against cardiovascular diseases [13,14].

Here, we focus on the effects of adiponectin on glucose and lipid metabolism and on its anti-atherosclerotic properties. Furthermore, we discuss the factors which increase circulating adiponectin levels.

## 2. Effects of Adiponectin on Glucose Metabolism

### 2.1. Possible Mechnisms for the Improvement of Glucose Metabolism by Adiponectin

#### 2.1.1. Reduction of Inflammation and Oxidative Stress and Improvement of Insulin Resistance by Adiponectin

The adipose tissue is an active endocrine organ that secretes a variety of hormones known as adipokines. Adipokines are secreted into the circulation and participate in the regulation of insulin sensitivity and glucose and lipid metabolism [15]. In obesity and metabolic syndrome, a highly inflammatory status is induced by the infiltration of inflammatory cells into the adipose tissue, especially activated macrophages. Under these conditions, the adipose tissue produces proinflammatory adipokines, such as tumor necrosis factor-alpha (TNF-α), interleukin (IL)-6, monocyte chemoattractant protein-1 (MCP-1), lipocalin-2, and resistin, which induce atherosclerosis [16]. In these circumstances, the production of adiponectin is markedly reduced. 

Chronically elevated levels of inflammatory cytokines could directly enhance insulin resistance and lead to disrupted insulin sensitivity, in turn impairing glucose and lipid metabolism. Epidemiological studies have reported that levels of pro-inflammatory and inflammatory cytokines such as C-reactive protein (CRP), TNF-α, IL-1β, and IL-6 were elevated in patients with type 2 diabetes and were associated with the development of type 2 diabetes [3,17,18,19,20,21,22,23,24,25]. 

Adiponectin could reduce inflammatory reactions [2,3], which may be associated with an improvement of insulin resistance. The anti-inflammatory properties of adiponectin are likely to be the major component of its beneficial effects for alleviating insulin resistance and vascular diseases [26]. Adiponectin has been reported to decrease CRP mRNA and protein [27] and inhibit the stimulation of nuclear factor-κB (NF-κB) signaling and TNF-α secretion from macrophages [28]. Adiponectin suppresses TNF-α-induced monocyte adhesion to human aortic endothelial cells and the expression of certain adhesion molecules [29]. Further, adiponectin reduces the expression of cell adhesion molecules and the activation of IL-8 and NFκB by decreasing TNF-α in endothelial cells [27,30]. Adiponectin also modulates macrophage function and phenotype [31]. IL-10 stimulation and IL-1 receptor antagonists are associated with the anti-inflammatory actions of adiponectin in human monocytes, monocyte-derived macrophages, and dendritic cells [32].

Adiponectin suppresses the generation of oxidative and nitrative stress by inhibiting inducible nitric oxide synthase (iNOS) and suppressing the expression of a nicotinamide adenine dinucleotide phosphate (NADPH) oxidase subunit [33], which improves insulin resistance. Further, adiponectin improves insulin resistance in the liver and skeletal muscle via adenosine monophosphate-activated protein kinase (AMPK) and peroxisome proliferator-activated receptor-α (PPAR-α) activation [34].

#### 2.1.2. Pancreatic β Cell Protective Effect of Adiponectin

Adiponectin prevents ceramide- or inflammatory cytokine-induced apoptosis in cultured β cells [35,36] and maintains β cell mass and glucose homeostasis in *ob/ob* mice and in a mice model of type 1 diabetes [35,37]. Adiponectin-null mice are more susceptible to caspase-8-induced β cell apoptosis [36]. Via adiponectin receptors AdipoR1 and AdipoR2, adiponectin stimulates the de-acylation of ceramide, yielding sphingosine after conversion to sphingosine 1-phosphate (S1P) by sphingosine kinase. The resulting conversion from ceramide to S1P promotes the survival of functional β-cell mass [38].

#### 2.1.3. Increase of Glucose Utilization and Fatty Acid Oxidation in Skeletal Muscles by Adiponectin

Adiponectin has been reported to improve glucose utilization and fatty acid (FA) oxidation in myocytes [39]. In addition, in mice fed with high fat/sucrose diet, adiponectin showed to increase energy expenditure by increasing FA oxidation and to increase glucose uptake in skeletal muscle [40]. Adiponectin increased glucose transporter-4 (GLUT-4) translocation and glucose uptake by rat skeletal muscle cells [41]. These beneficial effects of adiponectin on glucose metabolism were mainly via the activation of AMPK in skeletal muscles [42]. In addition, it has been suggested that adiponectin decreases insulin resistance by decreasing the muscular lipid content in obese mice [43].

#### 2.1.4. Adiponectin Reduces Hepatic Glucose Production

In the liver, adiponectin improves hepatic and systemic insulin resistance through the activation of AMPK and PPAR-α pathways [34]. Adiponectin has been reported to suppress both glycogenolysis and gluconeogenesis [42] by reducing the rate-limiting enzymes for hepatic glucose production, such as glucose-6-phosphatase (G6Pase) and phosphoenolpyruvate carboxy kinase (PEPCK) [39,44,45,46,47]. Besides the suppression of G6Pase and PEPCK, adiponectin can suppress glucose production by reducing the availability of gluconeogenic substrates [47]. Adiponectin stimulates FA oxidation, which reduces gluconeogenic availability. 

#### 2.1.5. Adiponectin Increases Insulin-Stimulated Glucose Uptake by Adipocytes

Adiponectin treatment enhances insulin-stimulated glucose uptake via activation of AMPK in primary rat adipocytes [48]. Adiponectin directly targets insulin receptor substrate-1 (IRS-1) rather than the insulin receptor (IR) [49]. IRS-1 plays a crucial role in insulin mediation of glucose uptake in adipocytes [50]. Decreased levels of IRS-1 are significantly associated with insulin resistance and type 2 diabetes [51,52].

#### 2.1.6. Summary of Anti-Diabetic Effects of Adiponectin

Possible mechanisms for the improvement of glucose metabolism by adiponectin are shown in Figure 1.

### 2.2. Adiponectin and Development of Type 2 Diabetes

In a case–control series which was performed in the Pima Indian population [53], at baseline, the serum adiponectin level was significantly lower in the cases (*n* = 70) than in the controls (*n* = 70), and individuals who showed high serum adiponectin levels were less likely to develop type 2 diabetes than individuals with low serum adiponectin levels (incidence rate ratio 0.63 (95% confidence intervals (CI) 0.43–0.92); *p* = 0.02) [54]. In the population-based Monitoring of Trends and Determinants in Cardiovascular Disease (MONICA)/Cooperative Health Research in the Region of Augsburg (KORA) cohort study between 1984 and 1995 with follow-up until 2002 (mean follow-up 10.9 ± 4.7 years) [55], low levels of adiponectin were associated with an increased type 2 diabetes risk. The multivariable adjusted hazard ratio (HR) with 95% CI comparing tertile extremes was 2.65 (1.88-3.76) for adiponectin (bottom vs. top tertile), respectively [54]. A systematic review and meta-analysis of prospective studies was conducted to assess the association of serum adiponectin level with risk of type 2 diabetes. This meta-analysis included 19 studies, comprising a total of 39,136 participants and 7924 cases, and showed that type 2 diabetes risk was strongly associated with low levels of adiponectin [55]. Furthermore, other observational studies showed that low levels of adiponectin are significantly associated with the development of type 2 diabetes [23,25,56,57,58].

## 3. Effects of Adiponectin on Lipid Metabolism

### 3.1. Possible Mechanisms for the Improvement of Lipid Metabolism by Adiponectin

Adiponectin has been found to be correlated with various parameters of lipoprotein metabolism and, especially, it is associated with the metabolism of high-density lipoprotein (HDL) and TG. Adiponectin appears to induce an increase in serum HDL and, in addition, it lowers serum TG through the enhanced catabolism of TG-rich lipoproteins [59]. 

#### 3.1.1. Possible Mechanism for the Increase of HDL by Adiponectin

Almost all of the previous studies reported that serum adiponectin is positively correlated with serum HDL-C level [60,61,62,63,64,65,66]. Especially, HDL-C has been shown to be positively correlated with HMW adiponectin, which is considered the most biologically active form of adiponectin [60,64,65], independently of adiposity and of insulin sensitivity [61,63,64,65,67,68,69]. We also found that adiponectin was independently and positively correlated with HDL-C in 174 subjects without diabetes [70].

Adiponectin has been shown to increase HDL-C via an increase in the hepatic production of apo-AI, which is the major apolipoprotein of HDL, and through an increase in the production of ATP-binding cassette transporter A1 (ABCA1), which induces HDL assembly through reverse cholesterol transport [71,72,73,74,75,76]. Adiponectin has been shown to enhance ABCA1 expression through the activation of nuclear receptors including liver X receptor α and PPAR-γ [75]. 

Adiponectin-induced increase in HDL-C involves the down-regulation of hepatic lipase (HL) activity, given the reported inverse association of serum adiponectin with HL activity, which appears to be independent of measures of adiposity and insulin resistance [77,78].

Another possible mechanism underlying the adiponectin-induced up-regulation of HDL-C is the activation of lipoprotein lipase (LPL) by adiponectin and/or the improvement of insulin resistance, which can also reduce TG. 

#### 3.1.2. Possible Mechanisms for TG reduction by Adiponectin

The majority of previous studies have demonstrated a negative association between circulating adiponectin and serum TG [60,61,63,64,65,66,67,68,69,78]. Very low density lipoprotein (VLDL), one of TG-rich lipoproteins, has been found to be correlated with serum HMW adiponectin [66,79,80,81]. We also found that VLDL-C levels were inversely correlated with adiponectin levels independently of age, body mass index (BMI), gender, and glycemic control in patients with type 2 diabetes [82] The reported association of circulating adiponectin with VLDL apoB100 fractional catabolic rate suggests that the regulation of serum VLDL-C by adiponectin may involve VLDL catabolism [79,83,84]. A plausible explanation for the adiponectin-induced increase in TG catabolism is the regulation of LPL activity by adiponectin. It is well known that LPL, which is translocated to the endothelial cell surface of the vessels of heart, muscles, and adipose tissue, hydrolyses TG in TG-rich lipoproteins including chylomicrons and VLDL [85]. Serum adiponectin has been reported to be positively correlated with post-heparin LPL concentration and activity in the fasting state, apparently independently of insulin resistance and inflammation [86,87]. As mice over-expressing adiponectin display increased LPL gene expression and LPL activity in skeletal muscle during fasting and in adipose tissue mainly during the well-fed state [74,88], adiponectin may have a direct role in inducing LPL expression and activation in both skeletal muscle and adipose tissue. 

Another possible mechanism for TG reduction by adiponectin would be attributable to adiponectin-induced decrease in serum apo-CIII, a well-known inhibitor of LPL, as indicated by the reported negative association between circulating adiponectin and serum apo-CIII [89,90], and the down-regulation of apo-CIII-mRNA levels in adiponectin-treated human HepG2 hepatocytes [87]. In addition, the other mechanism of the adiponectin-induced up-regulation of VLDL catabolism involves the increased expression of VLDL receptor (VLDL-R) in skeletal muscle. Using adenovirus-mediated gene transduction, an increase of VLDL-R expression has been observed in adiponectin-treated myotubes, with an acute elevation of plasma adiponectin leading to the increased VLDL catabolism [87]. 

Insulin resistance increases activity and expression of hormone-sensitive lipase (HSL) in adipose tissue, which catalyzes the breakdown of TG, releasing free fatty acids (FFA) [91]. Increased FFA enter the liver and enhance the production of VLDL. Therefore, an improvement of insulin resistance by adiponectin may reduce HSL activity and result in a reduction of VLDL production. 

Adiponectin has been also shown to be associated with apo-B48, which is an apolipoprotein of chylomicrons from the small intestine [89,90]. A plausible explanation for this relationship is the up-regulation of postprandial TG catabolism by adiponectin, as indicated by the reported association of circulating adiponectin with heparin-releasable LPL activity in subcutaneous adipose tissue, observed after a meal [77].

#### 3.1.3. Effects of Adiponectin on LDL and Other Atherogenic Lipids

With regard to possible relationship between circulating adiponectin and low-density lipoprotein cholesterol (LDL-C), the majority of studies have shown no association [64,65,70,92,93,94]. Small dense LDL (sd-LDL) is considered an emerging risk factor for cardiovascular diseases (CVD). High sd-LDL levels have been reported to be associated with elevated TG levels and low HDL-C levels and constitute a common feature of type 2 diabetes and metabolic syndrome [95,96]. Oxidative modifications of LDL represent an early stage of atherosclerosis, and sd-LDL are more susceptible to oxidation than larger, more buoyant particles [96]. Adiponectin-mediated improvement of TG and HDL may reduce the atherogenic lipoprotein sd-LDL. Remnant lipoproteins, derived from VLDL and chylomicrons, have been considered to be atherogenic [97]. In patients with hypertriglyceridemia, TG-rich lipoproteins mainly increase during fasting and the postprandial state. Remnant lipoproteins directly and indirectly correlate to the enhancement of atherogenicity [98]. Therefore, the reduction of remnant lipoproteins due to the decrease of TG by adiponectin may contribute to the anti-atherogenic effects of adiponectin.

#### 3.1.4. Summary of Mechanisms for the Improvement of Lipid Metabolism by Adiponectin.

Possible mechanisms for the improvement of lipid metabolism by adiponectin are shown in Figure 2.

Adiponectin increases HDL-C via an increase in the hepatic production of apo-AI, through an increase in the expression of ABCA1 in peripheral tissues. Down-regulation of HL activity by adiponectin may also increase HDL-C. Increased LPL expression and activity in skeletal muscle and adipose tissue contribute to the reduction of TG-rich lipoproteins and the elevation of HDL. Adiponectin-induced decrease of hepatic apo-CIII production and adiponectin-induced up-regulation of VLDL-R in skeletal muscle also lead to the decrease of TG. An improvement of insulin resistance by adiponectin may reduce HSL activity and result in the reduction of VLDL production due to a decreased release of FFA from the adipose tissue to the liver.

## 4. Anti-Atherosclerotic Effects of Adiponectin

### 4.1. Improvement of Endothelial Function and Interaction Between Monocyte and Endothelium by Adiponectin

There is a close relationship between hypoadiponectinemia and peripheral arterial dysfunction [99,100,101]. Adiponectin knockout mice showed significantly increased neointimal hyperplasia, disordered endothelium-dependent vasodilation, and increased blood pressure, compared with wild-type mice [99,102,103]. Flow-mediated dilation of the brachial artery has a significant relationship with plasma HMW adiponectin levels in young healthy men [104]. The biosynthesis of nitric oxide (NO) is performed by AMPK and is mediated by adiponectin-induced phosphorylation of endothelial nitric oxide synthase (eNOS). Adiponectin inhibits the interaction between leukocytes and endothelial cells by reducing E-selectin and vascular cell adhesion molecule-1 induced by TNF-α, resistin, and IL-8, and by increasing endothelial NO [105], which results in the attenuation of monocyte attachment to endothelial cells [31]. Serum adiponectin concentration also showed a significant negative correlation with serum MCP-1 concentration (*r* = −0.244, *p* = 0.05) in postmenopausal women [106]. Adiponectin also reduces irregular high glucose-induced apoptosis and oxidative stress in human umbilical vein endothelial cells [105,107]. 

Elevated serum TG levels are an independent predictor of endothelial dysfunction. Lowering circulating TG levels by adiponectin may improve the endothelial function [108]. The increase of TG and decrease of HDL reduce the activity and expression of eNOS and disrupt the integrity of the vascular endothelium due to oxidative stress [109]. Diabetes-induced endothelial dysfunction is a critical and initiating factor in the genesis of diabetic vascular complications [110]. Therefore, reduction of TG, elevation of HDL, and improvement of glucose metabolism may ameliorate the endothelial function. 

### 4.2. Inhibition of Smooth Muscle Proliferation by Adiponectin

Rapid proliferation and migration of vascular smooth muscle cells (SMCs) toward the intima contribute to intimal thickening of arteries and atherosclerosis development. Adiponectin blocks the proliferation and migration of human aortic SMCs by inhibiting several atherogenic growth factors, including platelet-derived growth factor, basic fibroblast growth factor, and heparin-binding epidermal growth factor [111,112].

### 4.3. Increase of Macrophage Cholesterol Efflux and Suppression of Foam Cell Formation

Serum HDL-C levels are inversely correlated to the risk of atherosclerotic cardiovascular diseases. The reverse cholesterol transport is one of the major protective systems against atherosclerosis, in which HDL particles play a crucial role, carrying cholesterol derived from peripheral tissues to the liver. ABCA1 receptors has been identified as important membrane receptors to generate HDL by cholesterol efflux from foam cells. Adiponectin has been reported to up-regulate the expression of ABCA1 in human macrophages and enhance apo-AI-mediated cholesterol efflux from macrophages [75]. Recently, Marsche et al. investigated the association between cholesterol efflux capacity and metabolic parameters in 683 participants (281 youths, of whom 227 were overweight/obese; 402 adults, of whom 197 were overweight/obese). They found that hypoadiponectinemia is a robust predictor of reduced cholesterol efflux capacity in adults, irrespective of BMI and fat distribution [113]. Adiponectin markedly suppressed foam cell formation in oxidized LDL-treated macrophages from diabetic subjects, which was mainly attributed to an increase in cholesterol efflux [114]. In addition, a deletion of adipoR1 in macrophages from diabetic patients accelerated foam cell formation induced by oxidized LDL [114]. A strong positive correlation was noted between decreased serum adiponectin and impaired cholesterol efflux capacity, both before and after adjustment for HDL-C and apo-AI in diabetic patients (both *p* < 0.001) [114]. The adiponectin-treated macrophages contained fewer lipid droplets stained by oil red O [3]. The adipocyte-derived plasma protein adiponectin suppressed macrophage-to-foam cell transformation by reducing the expression of class A macrophage scavenger receptor at both mRNA and protein levels [3]. 

Kubota et al. carried out serum cholesterol efflux studies in individuals with glucose intolerance [115]. An inverse correlation was found between the cholesterol efflux capability and the extent of glucose intolerance in an oral glucose tolerance test. An improvement of glucose metabolism and insulin resistance may ameliorate cholesterol efflux. Interestingly, enhanced cholesterol efflux to HDL through the ABCA1 transporter was observed in hypertriglyceridemic patients with type 2 diabetes [116,117]. Further, enhanced efflux of cholesterol from ABCA1-expressing macrophages to serum was observed in patients with hypertriglyceridemia [118]. 

### 4.4. Putative Molecular Anti-Atherosclerotic Effects of Adiponectin

Possible anti-atherosclerotic effects of adiponectin are shown in Figure 3.

## 5. How can We Increase Adiponectin?

### 5.1. Weight Loss

A systematic review which assessed the consequences of all types of obesity surgery showed that adiponectin was significantly increased after bariatric surgery [119]. Sibutramine is an anti-obesity medication whose effects on weight loss have been widely explored. A systematic review and meta-analysis of available evidence was conducted in order to calculate the effect size of sibutramine therapy on adipokines [120]. Random-effect meta-analysis evidenced a significant increase of adiponectin (weighted mean difference (WMD) 9.86%, 95%CI: 1.76, 17.96, *p* = 0.017) following sibutramine therapy. A systematic review and meta-analysis of clinical trials that assessed the effect of a low-calorie diet on adiponectin concentration showed that a weight-loss diet can substantially increase the overall adiponectin concentration (Hedges’ *g* = 0.34, 95% CI:0.17–0.50, *p* < 0.001) [121]. 

### 5.2. Exercise

We examined the effects of supervised aerobic exercise on serum adiponectin and lipids in patients with moderate dyslipidemia. In this study, 25 patients (mean BMI, 24.6 kg/m²; mean age, 39 years; mean total cholesterol, 226 mg/dL; mean TG, 149 mg/dL) without metabolic syndrome, diabetes, and hypertension underwent a 16-week supervised aerobic exercise program (60 min/day, 2 to 3 times/week) with moderate exercise intensity [122]. Adiponectin significantly increased by 51% at week 16, although changes in these parameters were not significant at week 8 [123]. Several meta-analyses have shown that the exercise increased serum adiponectin [124,125,126,127], supporting our study result.

### 5.3. Nutritional Factors

#### 5.3.1. Vitamins

Vitamin D has been proposed to have anti-inflammatory properties. A meta-analysis was performed to examine the effect of vitamin D supplementation on adipocytokines in patients with type 2 diabetes [128]. In the meta-analysis of 20 randomized controlled trials (RCTs) (*n* =  1270 participants), vitamin D-supplemented groups had lower levels of CRP and TNF α and higher levels of leptin compared with control groups. However, no differences were observed for adiponectin. Also another meta-analysis did not indicate a significant effect of vitamin D supplementation on serum adiponectin levels [123].

A meta-analysis assessed the effects of vitamin K supplementation on a homeostasis model assessment of insulin resistance (HOMA-IR), fasting plasma glucose and insulin, CRP, adiponectin, leptin, or IL-6 levels [129]. A total of eight trials involving 1077 participants met the inclusion criteria. Vitamin K supplementation did not affect insulin sensitivity as measured by HOMA-IR, fasting plasma glucose and insulin, CRP, adiponectin, leptin, and IL-6 levels.

#### 5.3.2. Polyphenols

Resveratrol is a non-flavonoid polyphenol that naturally occurs as phytoalexin. The shell and stem of *Vitis vinifera* L. (Vitaceae) are the richest sources of this compound. A variety of in vitro and in vivo studies suggested the effectiveness of resveratrol in diabetes [130]. A systematic review and a meta-analysis of available RCTs to elucidate the role of resveratrol supplementation on adipokines showed a significant change in serum adiponectin concentrations following resveratrol supplementation (WMD: 1.10 μg/mL, 95% CI: 0.88, 1.33, *p* < 0.001) [131].

#### 5.3.3. Carotenoids

Astaxanthin is a naturally occurring red pigmented carotenoid classified as a xanthophyll, found in microalgae and seafood such as salmon, trout, and shrimp. Astaxanthin as a bioactive compound has a potential role in the prevention of atherosclerosis and a beneficial effect on adiponectin levels [132]. We performed an RCT of astaxanthin analyzing metabolic parameters. Placebo-controlled astaxanthin administration at doses of 0, 6, 12, 18 mg/day for 12 weeks was randomly allocated to 61 non-obese subjects with fasting serum TG of 120-200 mg/dL and without diabetes and hypertension, aged 25–60 years. Serum adiponectin was increased by astaxanthin (12 and 18 mg/day), and changes in adiponectin correlated positively with HDL-C changes, independent of age and BMI [133].

Carotenoids have been implicated in the regulation of adipocyte metabolism. Canas et al. compared the effects of mixed-carotenoid supplementation (MCS, which contains β-carotene, α-carotene, lutein, zeaxanthin, lycopene, astaxanthin, and γ-tocopherol) to those of a placebo on adipokines in children with obesity [134]. An RCT to evaluate the effects of MCS over 6 months was performed. Twenty children (6 male and 14 female) with simple obesity (BMI > 90%) and a mean age (± SD) of 10.5 ± 0.4 years, were enrolled. MCS increased total adiponectin and HMW adiponectin compared with the placebo.

Another study assessed the effects of 280 mL of tomato juice (containing 32.5 mg of lycopene) consumed daily in addition to a normal diet and an exercise program for 2 months [135]. The tomato juice supplementation significantly reduced body weight, body fat, waist circumference, and BMI, and significantly increased serum adiponectin levels. The intervention included 10 weeks of consumption of a tomato-based diet (≥25 mg lycopene daily) with an intermediate 2-week washout and was performed in 70 postmenopausal women with mean age of 57.2 years and mean BMI of 30.0 kg/m^2^ [136]. After the tomato intervention, adiponectin concentration increased (ratio 1.09, 95%CI 1.00–1.18), with a stronger effect observed among nonobese women (ratio 1.13, 95% CI 1.02–1.25). 

A positive association between concentrations of β-carotene and adiponectin independent of sex, age, smoking status, BMI, and waist circumference was observed in non-diabetic obese subjects. [137]. In this cross-sectional study which assessed whether serum carotenoids are associated with HMW adiponectin in 437 Japanese subjects (116 men and 321 women), serum β-carotene concentrations were significantly associated with serum HMW adiponectin concentrations in both sexes (standardized β coefficient = 0.197, *p* = 0.036 for men; standardized β coefficient = 0.146, *p* = 0.012 for women) [138].

Serum β-cryptoxanthin levels are lower in overweight subjects than in normal subjects. An intervention study consisted of a three-week long before-and-after controlled trial, where β-cryptoxanthin (4.7 mg/day) was given to 17 moderately obese postmenopausal women [139]. Serum HMW adiponectin levels significantly increased after this intervention. An RCT tested the effects of antioxidant (AOX) supplementation (vitamin E, 800 IU/day; vitamin C, 500 mg/day; β-carotene, 10 mg/day) on insulin sensitivity and adipokines in overweight and normal-weight individuals (*n* = 48, aged 18–30 years) [140]. The participants received either AOX or a placebo for 8 weeks. Adiponectin increased in both AOX groups. In another RCT by the same research group, overweight (BMI, 33.2 ± 1.9 kg/m^2^) and comparative normal-weight (BMI, 21.9 ± 0.5 kg/m^2^) adults, aged 18 to 30 years old (*n* = 48), were enrolled [141]. Either daily AOX treatment or placebo were administered for 8 weeks to the study subjects who completed a standardized 30-minute cycle exercise bout at baseline and week 8. Adiponectin was increased in both overweight and normal-weight AOX groups (22.1% vs. 3.1%; *p* < 0.05) but reduced in placebo groups. 

#### 5.3.4. Omega-3 FA

Fish oil, a source of omega-3 FAs, improves insulin sensitivity in animal experiments, but findings remain inconsistent in humans. A meta-analysis of RCTs determined the effect of omega-3 FA consumption on circulating adiponectin in humans [142]. Fourteen RCT arms evaluated fish oil (fish oil, *n* = 682; placebo, *n* = 641). Fish oil increased adiponectin by 0.37 μg/mL (95% CI 0.07; 0.67, *p* = 0.02). To determine the effects of omega-3 FA supplementation on adipocytokine levels in adult prediabetic and diabetic individuals, a meta-analysis of RCTs was performed [143]. Fourteen individual studies (*n* = 685) were included in the meta-analysis. Omega-3 FA supplementation increased adiponectin by 0.48 μg/mL (95% CI, 0.27 to 0.68; *p* < 0.00001). In the meta-analysis of RCTs which assessed the effects of omega-3 FA in women with polycystic ovary syndrome (PCOS), nine trials involving 591 patients were included. Compared with the control group, omega-3 FA increased adiponectin level (weighted mean difference (WMD) 1.34; 95% CI 0.51 to 2.17; *p* = 0. 002) [144]. In the meta-analysis of RCTs in patients with type 2 diabetes, omega-3 FA increased adiponectin by 0.57 µg/mL (95% CI 0.15 to 1.31; *p* = 0.01) [145]. Another meta-analysis in patients with type 2 diabetes showed a nonsignificant increase (MD = 0.17 µg/mL (95% CI 0.11 to 0.44)) of adiponectin [146].

### 5.4. Anti-Diabetic Drugs

#### 5.4.1. Thiazolidinediones

A systematic review which summarizes the evidence of the effect of thiazolidinediones (pioglitazone and rosiglitazone) on circulating adiponectin levels was performed through a systematic search in PubMed, Scopus, and Cochrane Library. A significant increase in adiponectin (80-178%) after thiazolidinediones treatment was observed in all included studies [147]. Our systematic review also reported that pioglitazone increased serum adiponectin levels [148]. Further, stopping pioglitazone was associated with a subsequent decrease in adiponectin (from 9.7 ± 9.1 to 5.1 ± 4.5 μg/ml) [149].

#### 5.4.2. Metformin

To provide high-quality evidence about the effect of metformin on adipocytokines in patients with PCOS, relevant studies that assessed the levels of adiponectin in patients with PCOS treated with metformin were reviewed and analyzed [150]. A total of 34 data sets were included, with four different outcomes, involving 744 women with PCOS. Metformin treatment was associated with significantly elevated serum adiponectin concentrations [standard mean difference (SMD) −0.43; 95%CI −0.75 to −0.11]. In a meta-analysis to investigate and determine the role of metformin on serum adiponectin levels in patients with type 2 diabetes, 18 cohort studies conducted among Asians and Caucasians from 2004 to 2013 were examined [151]. Post-treatment serum adiponectin levels were higher than pre-treatment levels in patients with type 2 diabetes (SMD = 0.19, 95% CI 0.09 to 0.30, *p* < 0.001).

#### 5.4.3. α-Glycosidase Inhibitors

Miglitol, one of α-glycosidase inhibitors, has been reported to increase serum adiponectin levels [152]. Adiponectin levels were significantly increased by miglitol (*p* < 0.01), and the significant increase in adiponectin by miglitol was inversely correlated with the ratio between the 60 minute change in blood glucose at three months and the change at baseline (*r* = −0.59, *p* = 0.02), which was independent of age, sex, changes in hemoglobin A1c and BMI, and the baseline concentration of adiponectin [153]. Another α-glycosidase inhibitor, acarbose, has been also reported to lead to a significant increase of adiponectin [154,155,156].

#### 5.4.4. Dipeptidyl peptidase-4 inhibitors (DPP4i)

The PubMed, Embase, and Cochrane library databases were searched from inception to February 2016. RCTs evaluating DPP4i (sitagliptin and vildagliptin) versus placebo or an active control drug in type 2 diabetic patients, lasting ≥12 weeks, were identified [157]. Weighted mean differences in adiponectin levels were calculated by using a fixed- or random-effects model. Ten RCTs, including 1495 subjects, were identified. Compared with the placebo, DPP4i (sitagliptin and vildagliptin) treatment significantly elevated adiponectin levels by 0.74 μg/mL (95%CI, 0.45 to 1.03), whereas, the difference was 0.00 μg/mL (95% CI, −0.57 to 0.56) when using an active-comparison. 

#### 5.4.5. Glucagon-like peptide-1 (GLP-1) analogues

The GLP-1 receptor agonist liraglutide did not change adiponectin levels in women with PCOS [158]. Liraglutide reduced HbA1c and adiponectin (all *p* < 0.05) in patients with non-alcoholic steatohepatitis [159]. An eight-week liraglutide therapy was associated with an increase in the levels of adiponectin (4480 vs. 6290 pg/mL, *p* < 0.002) in patients with type 2 diabetes [160]. However, liraglutide reduced serum adiponectin levels in Japanese patients with type 2 diabetes [161,162]. Exenatide significantly increased adiponectin levels after three months compared with baseline in patients with obesity and type 2 diabetes (*p* < 0.05) [163]. The adiponectin level was significantly increased by the addition of exenatide (0.39 ± 0.32 vs. −1.62 ± 0.97 μg/mL in exenatide and placebo groups, respectively, *p* = 0.045) in patients with poorly controlled type 2 diabetes [164].

#### 5.4.6. Sodium–glucose cotransporter 2 inhibitors (SGLT-2i)

The new drugs for type 2 diabetes SGLT-2i are reversible inhibitor of SGLT-2, leading to a reduction of renal glucose reabsorption and a decrease of plasma glucose, in an insulin-independent manner. Since SGLT-2i are proved to be significantly associated with weight loss, we have predicted that SGLT-2 inhibitors may increase adiponectin [165]. Dapagliflozin, ipragliflozin, and canagliflozin showed a significant increase of adiponectin [166,167,168,169,170,171,172]. 

#### 5.4.7. Sulfonyl Urea

In RCTs which investigated the effects of new anti-diabetic drugs (pioglitazone, DPP4i, and SGLT-2i) on adiponectin, glimepiride, a sulfonyl urea, has been used as a comparator [168,173,174,175,176,177,178]. Glimepiride is less likely to increase adiponectin than other oral anti-diabetic drugs. To observe the efficacy and safety of adding glimepiride to an established insulin therapy in poorly controlled type 2 diabetes and to assess the resulting changes in the HMW adiponectin serum levels and glycemia after glimepiride treatment, 56 subjects with poorly controlled insulin-treated type 2 diabetes were randomly assigned to either the glimepiride-treated group (*n*  =  29) or the insulin-increasing group (*n*  =  27) [179]. HMW adiponectin serum levels were significantly increased in the glimepiride-treated group compared with the insulin-increasing group. Changes in HbA1c were inversely correlated with changes in serum HMW adiponectin in the glimepiride-treated group (*r* =  −0.452, *p* =  0.02).

### 5.5. Hypolipidemia Drugs

#### 5.5.1. Statin

A meta-analysis of 12 RCTs with 16 comparisons and 1042 patients showed that serum adiponectin was not significantly affected by simvastatin (WMD: 0.42 μg/mL; 95% CI, −0.66 to 1.50 μg/mL) [180]. In a systematic review and meta-analysis of 43 studies, a significant increase in plasma adiponectin levels was observed after statin therapy (WMD: 0.57 μg/mL, 95% CI: 0.18 to 0.95, *p* = 0.004) [181]. In subgroup analysis, atorvastatin, simvastatin, rosuvastatin, pravastatin, and pitavastatin were found to change plasma adiponectin concentrations by 0.70 μg/mL (95% CI: −0.26 to 1.65), 0.50 μg/mL (95% CI: −0.44 to 1.45), −0.70 μg/mL (95% CI: −1.08 to −0.33), 0.62 μg/mL (95% CI: −0.12 to 1.35), and 0.51 μg/mL (95% CI: 0.30 to 0.72), respectively.

#### 5.5.2. Ezetimibe

A meta-analysis of 23 RCTs did not suggest any significant effect of adding ezetimibe to statin therapy on plasma concentrations of adiponectin (SMD 0.34, 95% CI −0.28 to 0.96; *p* = 0.288) [182].

#### 5.5.3. Fibrate

Out of 12 RCTs comprising 443 cases and 437 controls met the selection criteria for systematic review, 9 RCTs (399 cases and 401 controls) were included in the meta-analysis. Quantitative data synthesis revealed a significant effect of fibrate therapy in increasing circulating adiponectin levels (WMD: 0.38 μg/mL; 95%CI: 0.13 to 0.63 μg/mL; *p* = 0.003) [183]. In the head-to-head comparison of fibrates versus statins for the elevation of circulating adiponectin concentrations by a systematic review and meta-analysis, monotherapies with either fibrates or statins had comparable effects on circulating concentrations of adiponectin [184].

### 5.6. Anti-Hypertensive Drugs

#### Angiotensin II receptor blocker (ARB)

Telmisartan has been proposed to be a promising cardiometabolic ARB due to its unique PPAR-γ-inducing property. In a meta-analysis of RCTs, the pooled analysis suggested a significant increase in % changes of adiponectin (0.75; 95% CI, 0.40 to 1.09; *p* < 0.0001) among patients with metabolic syndrome randomized to receive telmisartan or control therapy [185]. The pooled analysis of the 11 trials (1088 patients) demonstrated a statistically significant increase in the percent changes of adiponectin levels (MD, 15.74%; 95% CI, 4.95% to 26.52%; *p* = 0.004) with telmisartan relative to other ARB therapies [186]. A systematic review of the effect of telmisartan on insulin sensitivity in hypertensive patients with insulin resistance or diabetes was performed [187]. Eight trials involving a total of 763 patients met the inclusion criteria. Telmisartan was superior to other ARBs in increasing adiponectin level (MD, 0.93 μg/dL; 95% CI, 0.28 to 1.59 μg/dL; *p* = 0.005).

### 5.7. Summary of Possible Factors Which Increase Circulating Adiponectin Levels

The summary of possible factors which increase circulating adiponectin levels are shown in Table 1.

## 6. Conclusions

Adiponectin reduces inflammatory cytokines and oxidative stress, which lead to an improvement of insulin resistance. Adiponectin-induced improvement of insulin resistance and adiponectin itself reduce hepatic gluconeogenesis and glycogenolysis and increase the utilization of glucose and FA by skeletal muscles, resulting in lower glucose levels. Adiponectin has also β-cell protective effect. A great number of previous studies demonstrated that adiponectin increases HDL and decreases TG. Adiponectin increases ABCA1 and LPL and decreases hepatic lipase, which may elevate HDL. Increased mass and activity of LPL and VLDL-receptor and reduced apo-CIII may increase VLDL catabolism and result in the reduction of serum TG. Further, adiponectin has various anti-atherosclerotic properties such as reduction of scavenger receptor in macrophages and increase of cholesterol efflux. These findings suggest that high circulating adiponectin levels can protect against atherosclerosis. Weight loss, exercise, nutritional factors, anti-diabetic drugs, hypolipidemic drugs, and anti-hypertensive drugs have been associated with an increase of serum adiponectin levels.

## Figures and Tables

**Figure 1 ijms-20-01190-f001:**
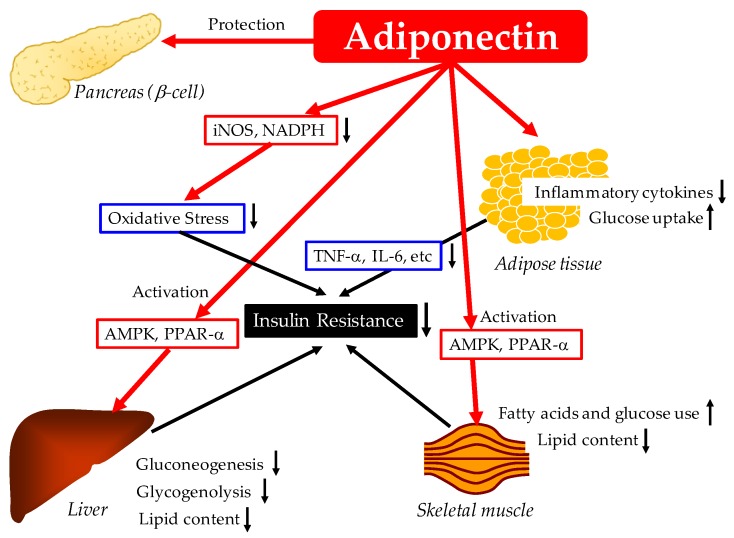
Possible mechanisms for the improvement of glucose metabolism by adiponectin. AMPK, adenosine monophosphate-activated protein kinase; IL-6, interleukin-6; iNOS, inducible nitric oxide synthase; NADPH, nicotinamide adenine dinucleotide phosphate; PPAR-α, peroxisome proliferator-activated receptor-α, TNF-α, tumor necrosis factor-α.

**Figure 2 ijms-20-01190-f002:**
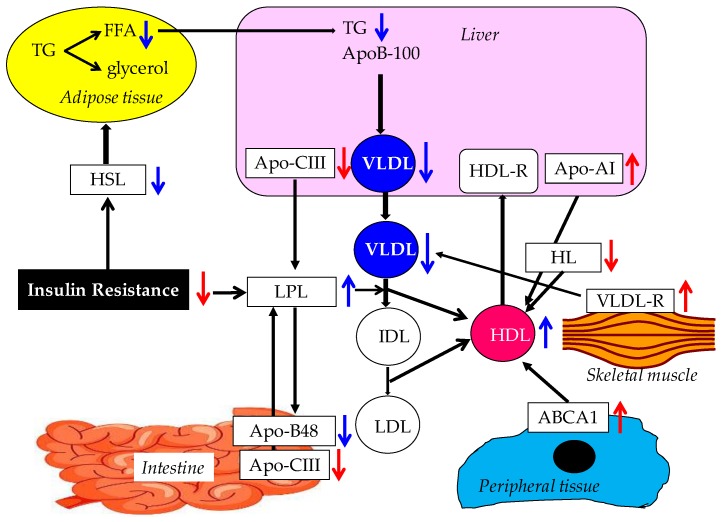
Possible mechanisms for the improvement of lipid metabolism by adiponectin. Red and blue arrows indicate direct and indirect lipid metabolism improving effects of adiponectin, respectively. ABCA1, ATP-binding cassette transporter A1; FFA, free fatty acids; HDL, high-density lipoprotein; HL, hepatic lipase; HSL, hormone-sensitive lipase; IDL, intermediate-density lipoprotein; LDL, low-density lipoprotein; TG, triglyceride; VLDL, very low density lipoprotein; VLDL-R, very low density lipoprotein-receptor.

**Figure 3 ijms-20-01190-f003:**
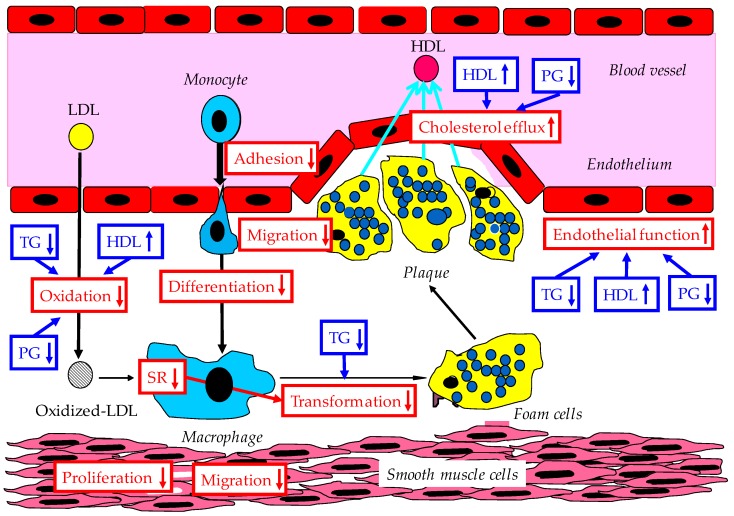
Possible anti-atherosclerotic effects of adiponectin and improvement of lipid/glucose metabolism by adiponectin. The red words in the red squares show the anti-arteriosclerotic effects of adiponectin. HDL, high-density lipoprotein; LDL, low-density lipoprotein; PG, plasma glucose; SR, scavenger receptor; TG, triglyceride.

**Table 1 ijms-20-01190-t001:** Possible factors which increase circulating adiponectin levels.

**1. Weight Loss**
Bariatric Surgery
Sibutramine
Low Calorie Diet
**2. Exercise**
**3. Nutritional Factors**
Resveratrol
Astaxanthin
Mixed-Carotenoid Supplementation (β-carotene, α-carotene,
Lutein, Zeaxanthin, Lycopene, Astaxanthin, γ-tocopherol)
Tomato Juice
β-carotene
β-cryptoxanthin
Antioxidant Supplementation (Vitamin E, Vitamin C, β-carotene)
Omega-3 Fatty Acids
**4. Anti-Diabetic Drugs**
Thiazolidinediones
Metformin
α-Glycosidase Inhibitors (Miglitol, Acarbose)
Dipeptidyl Peptidase-4 Inhibitors
Glucagon-Like Peptide-1Analugues (Liraglutide < Exenatide)
Sodium-Glucose Cotransporter 2 Inhibitors
**5. Hypolipidemia Drugs**
Statin
Fibrate
**6. Anti-Hypertensive Drugs**
Angiotensin II Receptor blockers (Telmisartan)

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
