# Peer review of "Beneficial Effects of Adiponectin on Glucose and Lipid Metabolism and Atherosclerotic Progression: Mechanisms and Perspectives"

_ijms, 2019, doi:10.3390/ijms20051190_

Reviewer 1 Report

In this review article, Prof. Yanai and Yoshida underscored the positive effects of adiponectin on glucose and lipid metabolism, and its anti-atherosclerotic properties and also discussed the factors which increase circulating adiponectin levels. The manuscript was relatively well prepared. I am overall supportive of this paper. Nevertheless, the authors should provide additional information to improve further the manuscript.

1. As adiponectin also acts locally on the adipocytes to increase glucose uptake, contributing to euglycemia, it is better to present or summarize the healthy effect of adiponectin on adipocyte glucose metabolism in Section 2 of the manuscript.

2. The Section 3 regarding the effects of adiponectin on lipid metabolism should be strengthened. In addition to discussing the effect of adiponectin on serum TG and lipoprotein (HDL and LDL), authors should include the benefits of adiponectin on lipid metabolism/accumulation in major peripheral metabolism sites including adipose, liver, and skeletal muscle tissues.

3. As authors use a large part of the manuscript to discuss the effect of adiponectin on glucose and lipid metabolism, it is plausible to incorporate this contents in the title where only anti-atherosclerotic properties of adiponectin was presented.

Also, it is necessary to discuss whether the positive effect of adiponectin on glucose and lipid metabolism contributes to its anti-atherosclerotic properties.

Author Response

List of Modifications (Manuscript ID ijms-451254)

Responses to comments of reviewers

Reviewer 1

Thank you for your valuable comments on our paper.

1.      According to “As adiponectin also acts locally on the adipocytes to increase glucose uptake, contributing to euglycemia, it is better to present or summarize the healthy effect of adiponectin on adipocyte glucose metabolism in Section 2 of the manuscript.”

We added the following sentences by citing more 4 references.

2.1.5. Adiponectin Increases Insulin-Stimulated Glucose Uptake by Adipocytes

Adiponectin treatment enhances insulin-stimulated glucose uptake via activation of AMPK, in primary rat adipocytes [49]. Adiponectin directly targets insulin receptor substrate-1 (IRS-1) rather than the insulin receptor (IR) [50]. The IRS-1 plays a crucial role in insulin mediation of glucose uptake in adipocytes [51]. Decreased IRS-1 is significantly associated with insulin resistance and type 2 diabetes [52,53]

2.      According to “The Section 3 regarding the effects of adiponectin on lipid metabolism should be strengthened. In addition to discussing the effect of adiponectin on serum TG and lipoprotein (HDL and LDL), authors should include the benefits of adiponectin on lipid metabolism/accumulation in major peripheral metabolism sites including adipose, liver, and skeletal muscle tissues.”

We added the following sentences.

Insulin resistance increases activity and expression of hormone-sensitive lipase (HSL) in adipose tissue, which catalyzes the breakdown of TG, releasing free fatty acids (FFA) [95]. Increased FFA enter liver and enhance production of VLDL. Therefore, an improvement of insulin resistance by adiponectin may reduce HSL activity and result in reduction of VLDL production.

Small dense LDL (sd-LDL) is considered an emerging risk factor for cardiovascular diseases (CVD). High sd-LDL levels have been reported to be associated with elevated TG levels and low HDL-C levels, constitutes a common feature of type 2 diabetes and metabolic syndrome [99,100]. Oxidative modifications of LDL represent an early stage of atherosclerosis, and sd-LDL are more susceptible to oxidation than larger, more buoyant particles [100]. Adiponectin-mediated improvement of TG and HDL may reduce atherogenic lipoprotein, sd-LDL. Remnant lipoproteins, derived from VLDL and chylomicrons, have been considered to be atherogenic [101]. In patients with hypertriglyceridemia, TG-rich lipoproteins mainly increase during fasting and the postprandial state. Remnant lipoproteins directly and indirectly correlate to the enhancement of atherogenicity [102]. Therefore, reduction of remnant lipoproteins due to decrease of TG by adiponectin may contribute to anti-atherogenic effects of adiponectin

Adiponectin increases HDL-C via an increase in the hepatic production of apo-AI, through an increase in expression of ABCA1 in peripheral tissues. Down-regulation of HL activity by adiponectin may also increase HDL-C. Increased LPL expression and activity in skeletal muscle and adipose tissue contribute to reduction of TG-rich lipoproteins and elevation of HDL. Adiponectin-induced decrease of hepatic apo-CIII production and adiponectin-induced up-regulation of VLDL-R in skeletal muscle also lead to decrease of TG. An improvement of insulin resistance by adiponectin may reduce HSL activity and result in reduction of VLDL production due to decrease of release of FFA from adipose tissue to liver.

3.      According to “As authors use a large part of the manuscript to discuss the effect of adiponectin on glucose and lipid metabolism, it is plausible to incorporate this contents in the title where only anti-atherosclerotic properties of adiponectin was presented.”

We changed the title from “Anti-Atherosclerotic Properties of Adiponectin: Mechanisms and Perspectives” to “Beneficial Effects of Adiponectin on Glucose and Lipid Metabolism, and Atherosclerotic Progression: Mechanisms and Perspectives”.

4.      According to “Also, it is necessary to discuss whether the positive effect of adiponectin on glucose and lipid metabolism contributes to its anti-atherosclerotic properties.”

We added the following sentences.

Small dense LDL (sd-LDL) is considered an emerging risk factor for cardiovascular diseases (CVD). High sd-LDL levels have been reported to be associated with elevated TG levels and low HDL-C levels, constitutes a common feature of type 2 diabetes and metabolic syndrome [99,100]. Oxidative modifications of LDL represent an early stage of atherosclerosis, and sd-LDL are more susceptible to oxidation than larger, more buoyant particles [100]. Adiponectin-mediated improvement of TG and HDL may reduce atherogenic lipoprotein, sd-LDL. Remnant lipoproteins, derived from VLDL and chylomicrons, have been considered to be atherogenic [101]. In patients with hypertriglyceridemia, TG-rich lipoproteins mainly increase during fasting and the postprandial state. Remnant lipoproteins directly and indirectly correlate to the enhancement of atherogenicity [102]. Therefore, reduction of remnant lipoproteins due to decrease of TG by adiponectin may contribute to anti-atherogenic effects of adiponectin.

Elevated serum TG levels are an independent predictor of endothelial dysfunction. Lowering circulating TG levels by adiponectin may improve endothelial function [112]. The increase of TG and decrease of HDL, reduce the activity and expression of eNOS and disrupts the integrity of vascular endothelium, due to oxidative stress [113]. Diabetes-induced endothelial dysfunction is a critical and initiating factor in the genesis of diabetic vascular complications [114]. Therefore, reduction of TG and elevation of HDL, and an improvement of glucose metabolism may improve endothelial function.

Kubota et al. carried out serum cholesterol efflux studies in individuals with glucose intolerance [121]. An inverse correlation was found between the cholesterol efflux capability and extent of glucose intolerance in an oral glucose tolerance test. An improvement of glucose metabolism and insulin resistance may ameliorate cholesterol efflux. Interestingly, enhanced cholesterol efflux to HDL through the ABCA1 transporter was observed in hypertriglyceridemic patients with type 2 diabetes [122,123]. Further, enhanced efflux of cholesterol from ABCA1-expressing macrophages to serum was observed in patients with hypertriglyceridemia [124].

We remade Figure 3 including effects of lipid and glucose improvement by adiponectin.

Reviewer 2 Report

This manuscript systematic describes the diverse biological function of adiponectin in different metabolic related organs. The references cover 23-year discovery of adiponectin in which 22% of references were published within 3 years. The manuscript could provide wide range and some novel knowledge for beginning readers. However, there are some points could be improved.
1. The title does not match the content of the text. The title seems to focus on anti-atherosclerotic effect of adiponectin; however, the weight of this part in the text is low. Unless, the authors could provide a figure or a paragraph that explains the link among atherosclerosis, glucose homeostasis and lipid metabolism.
2. Figure 1 and figure 2 could add more details, such as the molecules involving in the paths. Only using red and black arrows is quite confusing.
3. Figure 3 could add indirect (glucose and lipid metabolisms) effects of adiponectin on anti-atherosclerosis to make a better conclusion for the manuscript.

Author Response

List of Modifications (Manuscript ID ijms-451254)

Responses to comments of reviewers

Reviewer 2

Thank you for your valuable comments on our paper.

1.      According to “The title does not match the content of the text. The title seems to focus on anti-atherosclerotic effect of adiponectin; however, the weight of this part in the text is low. Unless, the authors could provide a figure or a paragraph that explains the link among atherosclerosis, glucose homeostasis and lipid metabolism.”

We changed the title from “Anti-Atherosclerotic Properties of Adiponectin: Mechanisms and Perspectives” to “Beneficial Effects of Adiponectin on Glucose and Lipid Metabolism, and Atherosclerotic Progression: Mechanisms and Perspectives”.

We added the following sentences.

Small dense LDL (sd-LDL) is considered an emerging risk factor for cardiovascular diseases (CVD). High sd-LDL levels have been reported to be associated with elevated TG levels and low HDL-C levels, constitutes a common feature of type 2 diabetes and metabolic syndrome [99,100]. Oxidative modifications of LDL represent an early stage of atherosclerosis, and sd-LDL are more susceptible to oxidation than larger, more buoyant particles [100]. Adiponectin-mediated improvement of TG and HDL may reduce atherogenic lipoprotein, sd-LDL. Remnant lipoproteins, derived from VLDL and chylomicrons, have been considered to be atherogenic [101]. In patients with hypertriglyceridemia, TG-rich lipoproteins mainly increase during fasting and the postprandial state. Remnant lipoproteins directly and indirectly correlate to the enhancement of atherogenicity [102]. Therefore, reduction of remnant lipoproteins due to decrease of TG by adiponectin may contribute to anti-atherogenic effects of adiponectin.

Elevated serum TG levels are an independent predictor of endothelial dysfunction. Lowering circulating TG levels by adiponectin may improve endothelial function [112]. The increase of TG and decrease of HDL, reduce the activity and expression of eNOS and disrupts the integrity of vascular endothelium, due to oxidative stress [113]. Diabetes-induced endothelial dysfunction is a critical and initiating factor in the genesis of diabetic vascular complications [114]. Therefore, reduction of TG and elevation of HDL, and an improvement of glucose metabolism may improve endothelial function.

Kubota et al. carried out serum cholesterol efflux studies in individuals with glucose intolerance [121]. An inverse correlation was found between the cholesterol efflux capability and extent of glucose intolerance in an oral glucose tolerance test. An improvement of glucose metabolism and insulin resistance may ameliorate cholesterol efflux. Interestingly, enhanced cholesterol efflux to HDL through the ABCA1 transporter was observed in hypertriglyceridemic patients with type 2 diabetes [122,123]. Further, enhanced efflux of cholesterol from ABCA1-expressing macrophages to serum was observed in patients with hypertriglyceridemia [124].

We remade Figure 3 including effects of lipid and glucose improvement by adiponectin.
2. According to “Figure 1 and figure 2 could add more details, such as the molecules involving in the paths. Only using red and black arrows is quite confusing.”

We changed Figure 1 and Figure 2 as the followings

Figure 1. Possible mechanisms for the improvement of glucose metabolism by adiponectin. AMPK,

adenosine monophosphate-activated protein kinase; IL-6, interleukin-6; iNOS, inducible nitric oxide

synthase; NADPH, nicotinamide adenine dinucleotide phosphate; PPAR-a, peroxisome

proliferator-activated receptor-a, TNF-α, tumor necrosis factor-α.

Figure 2. Possible mechanisms for improvement of lipid metabolism by adiponectin. Red and blue arrows indicate direct and indirect lipid metabolism improving effects of adiponectin, respectively. ABCA1, ATP-binding cassette transporter A1; FFA, free fatty acids; HDL, high-density lipoprotein; HL, hepatic lipase; HSL, hormone-sensitive lipase; IDL, intermediate-density lipoprotein; LDL, low-density lipoprotein; TG, triglyceride; VLDL, very low-density lipoprotein; VLDL-R, very low-density lipoprotein-receptor.
3. According to “Figure 3 could add indirect (glucose and lipid metabolisms) effects of adiponectin on anti-atherosclerosis to make a better conclusion for the manuscript.”

We changed Figure 3 as the following.

Figure 3. Possible anti-atherosclerotic effects of adiponectin and improvement of lipid/glucose metabolism by adiponectin. The red words in red squares show the anti-arteriosclerotic effects of adiponectin. HDL, high-density lipoprotein; LDL, low-density lipoprotein; PG, plasma glucose; SR, scavenger receptor; TG, triglyceride.